# Priority setting of vaccine introduction in Bangladesh: a multicriteria decision analysis study

Mohammad Sabbir Haider [1,2] Sitaporn Youngkong [1,3]
Montarat Thavorncharoensap [1,3] Praveen Thokala [4]

¹Mahidol University Health Technology Assessment (MUHTA) Graduate Program, Mahidol University, Bangkok, Thailand
²Directorate General of Health Services, Government of Bangladesh Ministry of Health and Family Welfare, Dhaka, Bangladesh
³Social and Administrative Pharmacy Division, Department of Pharmacy, Faculty of Pharmacy, Mahidol University, Bangkok, Thailand
⁴Health Economics and Decision Science, School of Health and Related Research (ScHARR), The University of Sheffield, Sheffield, UK

**Correspondence to**
Dr Sitaporn Youngkong;
sitaporn.you@mahidol.edu

## ABSTRACT

**Objective** To prioritise vaccines for introduction in Bangladesh.

**Methods** Multicriteria decision analysis (MCDA) process was used to prioritise potential vaccines for introduction in Bangladesh. A set of criteria were identified, weighted and assigned scores by relevant stakeholders (n=14) during workshop A. The performance matrix of the data of vaccines against the criteria set was constructed and validated with the experts (n=6) in workshop B. The vaccines were ranked and appraised by another group of stakeholders (n=10) in workshop C, and the final workshop D involved the dissemination of the findings to decision-makers (n=28).

**Results** Five criteria including incidence rate, case fatality rate, vaccine efficacy, size of the population at risk and type of population at risk were used quantitatively to evaluate and to score the vaccines. Two other criteria, cost-effectiveness and outbreak potentiality, were considered qualitatively. On deliberation, the Japanese encephalitis (JE) vaccine was ranked top to be recommended for introduction in Bangladesh.

**Conclusions** Based on the MCDA results, JE vaccine is planned to be recommended to the decision-makers for introduction into the national vaccine benefit package. The policymakers support the use of systematic evidence-based decision-making processes such as MCDA for vaccine introduction in Bangladesh, and to prioritise health interventions in the country.

## Strengths and limitations of this study

► Multicriteria decision analysis (MCDA) process was used to support vaccine introduction decision-making in Bangladesh, contributing to transparency and evidence-informed priority setting.
► Participation of a wide range of stakeholders in this MCDA study ensured the transparency and accountability of decision-making, which is essential for a fair priority setting process.
► Data on the vaccines on the different criteria were gathered from systematic evidence synthesis and validated with experts, and good practice MCDA guidance was followed to elicit the preferences and rank the list of vaccines.
► Different sets of stakeholders took part in the workshops, resulting in a lack of a consistent group of stakeholders (and hence values or preferences) throughout the MCDA process.
► Stakeholders from private sectors and representatives of patient groups were not involved in the process, leading to uncertainty in accountability of the results to those stakeholders.

## INTRODUCTION

Vaccination is the most effective public health measure to prevent infectious diseases.[1 2] Governments in developing countries prefer to invest in vaccination programmes that can be financially sustainable[3–5] and while countries often consider cost-effectiveness, this should not be the only criterion for the selection of interventions.[6 7] Different criteria, such as disease severity, effectiveness, accessibility, quality of care and equity, should be considered during healthcare priority setting.[8]

Decision-making around the introduction of new vaccines in the healthcare benefit package is complex.[9] There are systematic and evidence-based methods,[10] using priority setting to allocate the scarce resources to meet increasing demand.[11] Multicriteria decision analysis (MCDA) is one such approach which evaluates different options considering multiple criteria in an explicit manner,[12] to aid decision-makers to make rational decisions.[13] MCDA can be a useful approach to support inclusion of health interventions in the benefit package.[7]

Vaccine preventable diseases such as dengue, human papillomavirus (HPV), influenza, Japanese encephalitis (JE) and typhoid are prevalent in Bangladesh.[14–18] These diseases can be prevented by the introduction of new or underused vaccines by the government of Bangladesh. However, new vaccines have considerable budget impact and need to be prioritised for introduction into the benefit package.[19] In the past, decision-making for vaccine introduction was ad-hoc

but there is increasing interest in prioritisation using systematic evaluation of multiple criteria.[19]

As such, we conducted an MCDA study to support prioritisation of vaccines for introduction in the benefit package in Bangladesh. The aims of the study are to support prioritisation of health interventions using an evidence-based systematic process incorporating multiple criteria and involving key relevant stakeholders, and to provide national decision-makers with scientific recommendations on vaccine introduction to better use the limited resources in Bangladesh.

## METHODS

We followed the steps outlined in good practice guidelines for the use of MCDA in healthcare.[20 21] As stakeholder involvement is key, we conducted four workshops (between October 2019 and January 2020) with the relevant stakeholders during the MCDA process. The steps and the workshops are described in further detail below.

### Identifying the list of potential vaccines for introduction

The potential vaccines for prioritisation were identified from the recommendations of the WHO, Gavi the vaccine alliance, and Centers for Disease Control and Prevention in the USA (CDC-US). Vaccines which were currently in the expanded programme on immunisation (EPI) programme of the neighbouring countries were also considered. From these sources, vaccines that were not yet introduced in Bangladesh were identified as potential vaccines to be evaluated.

### Selecting criteria for vaccine introduction in Bangladesh

A three-step process was used to select criteria for vaccine introduction in Bangladesh. First, a systematic review was conducted to identify all potential criteria for vaccine introduction in Bangladesh, which is described elsewhere in detail.[22] Second, from this long list of criteria, a core team of three public health experts of Bangladesh (including the lead author, MSH) excluded criteria that cannot be quantified (eg, political will) and those that were mentioned less frequently.

Finally, the potential criteria list was ranked in workshop A (WS-A) in October 2019, to identify the key criteria to be used for vaccine prioritisation. Stakeholders (n=14) included paediatricians (n=1), public health experts (n=6), virologists (n=2), epidemiologists (n=4) and health economists (n=1). In terms of affiliation, these stakeholders (n=14) were from directorate office (n=4), technical institutes (n=4), non-government organisations (NGOs) (n=3), National Immunization Technical Advisory Group (NITAG) (n=2) and health professional associations (n=1). The criteria, along with their definitions, were presented to the stakeholders (online supplementary A) who were then asked to rank each criterion from '1 to 10', where '1' was the most preferable and '10' was the least preferable criterion. The ranked order of criteria was transformed into ranking weight using the rank order

centroid (ROC) method.[23] Criteria were ranked based on the mean ROC weight, and the stakeholders selected a set of criteria by consensus to be used in the prioritisation of vaccines.

### Weighting and scoring

In the same workshop (WS-A), the stakeholders weighted the criteria using direct rating methods. Stakeholders discussed and then agreed by consensus to assign points to each criterion on a scale of 0–100, where '0' depicted the least important, and '100' represented the most important. To calculate the weights, the points assigned for each criterion was normalised (ie, by dividing the points allocated to each criterion by the sum of points of all criteria) using Equation 1.[24 25]

$$w_i = p_i / \sum p_i \qquad (1)$$

where, $w_i$ is the normalised weight of criterion $i$; $i$ is the index of criterion; $p_i$ is the points allocated to each criterion.

For scoring, the levels of criteria were identified by the core team from literature review and expert opinion. These were presented to the stakeholders in WS-A, who then assigned scores to the levels in each criterion individually. The stakeholders then deliberated on these individual scores and assigned scores to each level of the criterion by consensus. The range of scores was between 0 and 1, where, '0' depicted the lowest score, and '1' represented the highest score.

### Gathering evidence

Data for the potential vaccines were collected from databases and reports from key organisations such as EPI, Communicable Disease Control of Directorate General of Health Services (CDC-DGHS), Institute of Epidemiology, Disease Control and Research and International Centre for Diarrhoeal Disease and Research, Bangladesh. A performance matrix was constructed, which presents data for each vaccine against the set of criteria. Then, workshop B (WS-B) was arranged in November 2019 to validate the data with a group of public health and vaccine experts in the country (n=6), that is, public health experts who were working in the disease surveillance (n=2), DGHS (n=2), Health Economics Unit (n=1) and NITAG (n=1). After reviewing and validation, they signed off on the performance matrix.

### Rank ordering the potential vaccines

The scores for the different levels from the WS-A were combined with the validated performance matrix from the WS-B to calculate the scores for each vaccine on the different criteria. Then, using the additive method[21] (see Equation 2),[26] the scores of each vaccine corresponding to the criteria level were multiplied by the weight of each criterion (from WS-A) to calculate the total scores of each potential vaccine. The vaccines were ranked based on the total scores of each vaccine, with the highest total score ranked top, and the next highest total second, and so on.

$$V_j = \sum C_{ij} * W_i \qquad (2)$$

where $V_j$ is the total value for alternative $j$; $C_{ij}$ is the score of alternative $j$ on criteria $i$; $W_i$ is the weight attached to criteria $i$.

### Appraising the rank of vaccines

Workshop C (WS-C) was conducted in December 2019 to appraise the vaccines. Stakeholders included the experts in the area of vaccination (n=10), that is, epidemiologists (n=2), virologists (n=3), infectious disease specialists (n=2), surveillance experts (n=1) and members of the vaccination policy programme (n=2). The performance matrix of the potential vaccines was provided in a paper-based format (online supplementary B) and the stakeholders were asked to assign the rank to the seven potential vaccines individually, where '1' was the most preferable vaccine and '7' was the least preferable vaccine. The mean rank of each vaccine was calculated from the ranks provided by each stakeholder, using the ROC method.[23]

The ranking analysis of vaccines retrieved from step 5 (based on findings from WS-A and WS-B) were then presented to the stakeholders, along with the evidence of the cost-effectiveness and outbreak potentiality of each vaccine. Stakeholders then considered all this information and deliberated to reach a consensus on a final ranking of vaccines.

### Application of vaccine prioritisation process in Bangladesh health system

A final workshop D was organised in January 2020 with the policymakers (n=28) working in vaccine decision-making, vaccination programme implementation, vaccine-related research and disease surveillance. The stakeholders were representatives from the ministry of health (n=12), the directorate office of health (n=9), development partners (n=2), health professional associations (n=2) and NGOs (n=3). This workshop involved the dissemination of the whole vaccine prioritisation process (including the selection of criteria, identification of vaccines and the MCDA methods), along with the findings.

### Patient and public involvement

Patients and the general public were not involved in this study.

## RESULTS

### The list of potential vaccines for introduction in Bangladesh

WHO recommended 23 vaccines for routine vaccination globally, while the CDC-US recommended 16 vaccines and Gavi the vaccine alliance provided support against 16 infectious diseases.[27–29] Bangladesh so far introduced 10 vaccines in their benefit package and two additional vaccines for the Haj pilgrimage travellers. Therefore, there were 11 vaccines not included yet in the Bangladesh health benefit package. After discussion among the core team and vaccine experts, four vaccines were excluded:

**Table 1** Selecting criteria based on ranking from the WS-A

| Criteria | Rank | |
| --- | --- | --- |
| | Using the mean of individuals | Consensus after deliberation |
| Incidence rate of disease* | 1 | 1 |
| Case fatality rate* | 2 | 2 |
| Vaccine efficacy* | 3 | 3 |
| Size of population at risk* | 5 | 4 |
| Type of population at risk* | 6 | 5 |
| Outbreak potentiality | 4 | 6 |
| Cost-effectiveness | 7 | 7 |
| Severity of disease | 8 | 8 |
| Global target | 9 | 9 |
| Equity | 10 | 10 |

*Criteria selected for vaccine prioritisation in Bangladesh.
WS-A, workshop A.

tick-borne encephalitis and yellow fever as Bangladesh lacked incidence data for these diseases, and varicella and hepatitis A virus vaccines as they were not included in the benefit package of the neighbouring countries. Seven vaccines (ie, cholera, dengue, typhoid, HPV, influenza, JE and rotavirus) were then selected for consideration in the priority setting process.

### Prioritisation criteria for vaccine introduction in Bangladesh

Sixty-seven criteria were identified in the systematic review, from which the core team identified 10 criteria as being potentially most relevant (table 1). Definitions of these 10 criteria were derived from the literature review.[30–32]

In the WS-A, stakeholders discussed the importance of each of these 10 criteria and justification for inclusion in the set of prioritisation criteria to be used for vaccine introduction in Bangladesh. Participants ranked individually first and after deliberation, consensus was achieved. Table 1 presents the mean of individual ranking using ROC method and the final consensus ranking. Based on these rankings, stakeholders selected the top five criteria for vaccine prioritisation in Bangladesh (ie, incidence rate, case fatality rate, vaccine efficacy, size of population at risk and type of population at risk). In addition to these five quantitative criteria, stakeholders also decided to include two qualitative criteria: 'outbreak potentiality' and 'cost-effectiveness'. These two criteria were not weighted or scored explicitly, but were used in deliberative discussions.

### Performance matrix

The data on the performance of each of seven vaccines against the prioritisation criteria are presented in table 2. The table presents data on the five quantitative criteria used for weighting and scoring, as well as the two qualitative criteria that were used in deliberative discussions.

## Table 2 Performance matrix with data of vaccines on the criteria (after validation in WS-B)

| Vaccine preventable disease | Incidence rate — Number of new cases per 100 000 population per year | Case fatality rate — Percentage of death among the cases in a year | Vaccine efficacy — Effectiveness of vaccine or reduction of diseases provided by vaccine (%) | Size of population at risk — Number of population at risk of getting infection per year (in millions) | Type of population at risk — Type of population needed to be vaccinated | Cost effectiveness* — Cost-effectiveness results from published literature | Outbreak potentiality‡ |
|---|---|---|---|---|---|---|---|
| Cholera[56–59] | 1640 | 3.0 | 53 | 15.175 | Under five children | Cost-effective | High |
| Dengue[60–62] | 3700 | 0.16† | 66 | 2.18† | Adult/high-risk | Very cost-effective | High |
| HPV[63–66] | 10.6 | 0.0115 | 95 | 1.56 | Woman | Highly cost-effective | Low |
| Influenza[58 67–71] | 10 200 | 0.088 | 63 | 15.5 | High risk | Cost-effective | Low |
| Japanese encephalitis[16 58 72–75] | 2.7 | 30.0 | 96.2 | 10.77 | High risk | Very cost-effective | Medium |
| Rotavirus[58 76–78] | 1080 | 0.0055 | 43 | 15.175 | Under five children | Very cost-effective | High† |
| Typhoid[58 79–83] | 280 | 0.30 | 81.60 | 15.175 | Under five children | Cost-effective | Medium |

*Not included in weighting and scoring, used in deliberative discussions in WS-C for final rankings. Judgements on cost-effectiveness were made from conclusions from published literature which evaluated the cost-effectivenss of these vaccines in Bangladesh or similar countries.
†Expert opinion.
‡Not included in weighting and scoring, used in deliberative discussions in WS-C for final rankings.
HPV, human papillomavirus; WS-B, workshop B; WS-C, workshop C.

## Table 3 Points allocated, and the calculated weights, for the criteria (from WS-A)

| Criteria | Points | Weight |
|---|---|---|
| Incidence rate | 100 | 0.26 |
| Case fatality rate | 85 | 0.22 |
| Vaccine efficacy | 80 | 0.21 |
| Size of population at risk | 75 | 0.19 |
| Type of population at risk | 50 | 0.13 |

WS-A, workshop A.

It should be noted that expert opinion (from WS-B) was used when there was no data available from published literature.

As shown in table 2, influenza and dengue fever have the highest incidence among adults or high-risk groups but with relatively low case fatality rates. JE, on the other hand, has a relatively low incidence but with high case fatality rate (almost a third of patients dying from the condition). Among children, cholera and rotavirus seem to be with the highest incidence and cholera with a mortality rate of 3%. Vaccine efficacy seems to be excellent for JE and HPV (both above 90%), quite good for typhoid (above 80%), moderate for dengue and influenza (around 65%) and average for cholera (53%) and rotavirus (43%). All the vaccines seemed to be cost-effective or highly cost-effective. Finally, outbreak potential seems high for dengue, cholera and rotavirus.

### Weighting and scoring

The participants of the WS-A consensually assigned 100 points to the criterion of 'incidence rate' and four other criteria were assigned points in accordance, with the least important criterion, 'type of population at risk' assigned 50 points. The weight of each criterion was calculated by using the normalisation method, and the weight of 'incidence rate' was estimated as 0.26, as presented in table 3. 'Case fatality rate' and 'vaccine efficacy' were weighted similarly (0.22 and 0.21, respectively), 'size of the population at risk' had a weight of 0.19 and 'type of population at risk' had the lowest weight (0.13).

In the same workshop (WS-A), the stakeholders assigned scores for the different levels of the five criteria by consensus, using direct rating methods. For continuous criteria such as 'incidence rate', 'case fatality rate', 'vaccine efficacy' and 'size of the population at risk', the scores were assigned based on the levels of measures (eg, scores of 1, 0.8 and 0.55 for three levels for vaccine efficacy based on whether efficacy is >80%, 60%–80% or <60%), while the scores for categorical criteria such as 'type of population at risk' were based on the categories (eg, scores of 1, 0.8, 0.7 and 0.5 for children, high-risk groups, women and adults, respectively). The scores for the different levels of each criterion are presented in table 4.

**Table 4** Scores for the levels of criteria (from WS-A)

| Criteria | Levels | Score |
|---|---|---|
| Incidence rate | Level 1: >1000/100 000 | 1.0 |
| | Level 2: 100–1000/100 000 | 0.8 |
| | Level 3: 10–100/100 000 | 0.5 |
| | Level 4: <10/100 000 | 0.3 |
| Case fatality rate | Level 1 >10% | 1.0 |
| | Level 2: 1%–10% | 0.8 |
| | Level 3: <1% | 0.4 |
| Vaccine efficacy | Level 1: >80% | 1.0 |
| | Level 2: 60%–79% | 0.8 |
| | Level 3: <60% | 0.55 |
| Size of population at risk | Level 1: >10 million | 1.0 |
| | Level 2: 1–10 million | 0.8 |
| | Level 3: 100 000–1 million | 0.5 |
| | Level 4: <100 000 | 0.3 |
| Type of population at risk | Level A: children (<5 years) | 1.0 |
| | Level C: high-risk group | 0.8 |
| | Level B: women | 0.7 |
| | Level D: adult | 0.5 |

WS-A, workshop A.

### Rank ordering the potential vaccines

After combining the findings from tables 2–4 to estimate the score and weights (ie, the weights from WS-A, and the scores by combining the different levels from WS-A with the data from performance matrix validated in WS-B), the core team performed analysis of seven vaccines and produced the ranking results, as shown in table 5. Cholera vaccine was top-ranked with the highest total score of 0.86 primarily because it affects children, has a high incidence rate, high case fatality rate and with high size of population at risk. Despite having effective vaccines, JE and HPV ranked bottom (with scores of 0.74 and 0.68, respectively) because they have a low incidence rate and low size of population at risk.

### Appraising the rank of vaccines

In the WS-C, the stakeholders reviewed the performance matrix and each stakeholder ranked the vaccines individually first. The mean of their individual rankings are presented in table 6. Based on the deliberations of performance matrix, the stakeholders in WS-C ranked HPV, JE and rotavirus, as the first, second and third, respectively. The stakeholders discussed and highlighted the importance of the vaccine for women, which was why HPV was ranked as the first. Then, they gave priority to vaccines with high incidence rate and high case fatality rate; therefore, JE and rotavirus vaccines were ranked next highest. This contrasts with the findings from the quantitative MCDA exercise by the core team (see table 5 using

findings from WS-A and WS-B), which suggested cholera, typhoid and influenza as the top three ranking vaccines.

The results of ranking by the core team (table 5) were then presented to the stakeholders in WS-C, along with the information on the potentiality of outbreak of the diseases and cost-effectiveness (see table 2). After considering all this information, the stakeholders adjusted the ranking by consensus and the final ranking is presented in table 6. HPV, JE and rotavirus still remained top three but the ranking order changed with JE, HPV and rotavirus being first, second and third, respectively.

### Application of vaccine prioritisation process in Bangladesh health system

After dissemination of the findings, the policymakers agreed on the importance of appraising new interventions scientifically and supported the use of MCDA in the priority setting process for vaccine introduction decision-making. The key personnel of the ministry of health and family welfare, Bangladesh, stated that '*It is better for Bangladesh at present to have this system to prioritise vaccines in the country. Bangladesh, a lower-middle income country is graduating Gavi funding. So, we have to change our decision-making process from donor influenced decision-making to self-decision-making*'. Based on the MCDA results, JE vaccine is planned to be recommended to the decision-makers for introduction into the national vaccine benefit package. They also highlighted that after the selection of vaccines, the country should prepare for vaccine logistics such as cold-chain capacity and other programmatic issues.

## DISCUSSION
### Summary of the study

This study represents the first time an explicit priority setting process based on MCDA was used for the prioritisation of vaccines in Bangladesh. Vaccines selected for prioritisation were those which were recommended by the international organisations but not included in health benefit package of Bangladesh. The potential multiple criteria were identified systematically from published literature, and shortlisted in two phases to select five quantitative criteria and two qualitative criteria for the evaluation of the vaccines. Weighting and scoring of the quantitative criteria were explicit and participatory, and the tools used for eliciting scores and weights were user friendly and well understood by the stakeholders. The final ranking of the vaccines was determined after deliberative discussions based on the performance matrix, which considered both quantitative criteria and qualitative criteria.

### Statement of the principal findings

Through this explicit MCDA approach, JE vaccine was placed as the top-ranked vaccine and is planned to be recommended to the decision-makers for introduction into the national vaccine benefit package. The policymakers support the use of systematic evidence-based decision-making processes such as MCDA for vaccine

**Table 5** Rank order of vaccine using only quantitative criteria (from WS-A and WS-B)

| | Incidence rate | Case fatality rate | Vaccine efficacy | Size of population at risk | Type of population at risk | Total | |
|---|---|---|---|---|---|---|---|
| Weight of criteria | 0.26 | 0.22 | 0.21 | 0.19 | 0.13 | | |
| Levels | L1 L2 L3 L4 | L1 L2 L3 | L1 L2 L3 | L1 L2 L3 L4 | L-A L-B L-C L-D | Sum | Rank |
| Score of levels | 1.0 0.8 0.5 0.3 | 1.0 0.8 0.4 | 1.0 0.8 0.55 | 1.0 0.8 0.5 0.3 | 1.0 0.8 0.7 0.5 | | |
| Cholera | (0.26×1.0) 0.26 | (0.22×0.8) 0.17 | (0.21×0.55) 0.11 | (0.19×1.0) 0.19 | (0.13×1.0) 0.13 | 0.86 | 1 |
| Typhoid | (0.26×0.8) 0.20 | (0.22×0.4) 0.09 | (0.21×1.0) 0.21 | (0.19×1.0) 0.19 | (0.13×1.0) 0.13 | 0.82 | 2 |
| Influenza | (0.26×1.0) 0.26 | (0.22×0.4) 0.09 | (0.21×0.8) 0.16 | (0.19×1.0) 0.19 | (0.13×0.7) 0.09 | 0.79 | 3 |
| Rotavirus | (0.26×1.0) 0.26 | (0.22×0.4) 0.09 | (0.21×0.55) 0.11 | (0.19×1.0) 0.19 | (0.13×1.0) 0.13 | 0.78 | 4 |
| Dengue | (0.26×1.0) 0.26 | (0.22×0.4) 0.09 | (0.21×0.8) 0.16 | (0.19×0.8) 0.15 | (0.13×0.7) 0.09 | 0.75 | 5 |
| Japanese encephalitis | (0.26×0.3) 0.08 | (0.22×1.0) 0.22 | (0.21×1.0) 0.21 | (0.19×0.8) 0.15 | (0.13×0.7) 0.09 | 0.74 | 6 |
| HPV | (0.26×0.5) 0.13 | (0.22×0.4) 0.09 | (0.21×1.0) 0.21 | (0.19×0.8) 0.15 | (0.13×0.8) 0.10 | 0.68 | 7 |

*Data from performance matrix (table 2) were combined with the scores for different levels (table 4) to estimate the scores for each vaccine. These were then multiplied with weights (table 3) to calculate overall scores, which were then used for ranking.
HPV, human papillomavirus; WS-A, workshop A; WS-B, workshop B.

**Table 6** Ranking of vaccines

| Vaccine | Ranking from WS-C | Ranking from the analysis of WS-A and WS-B | Final ranking after deliberation in WS-C* |
|---|---|---|---|
| Japanese encephalitis | 2 | 6 | 1 |
| HPV | 1 | 7 | 2 |
| Rotavirus | 3 | 4 | 3 |
| Cholera | 5 | 1 | 4 |
| Typhoid | 4 | 2 | 5 |
| Dengue | 7 | 5 | 6 |
| Influenza | 6 | 3 | 7 |

*including consideration of information on cost-effectiveness and outbreak potential.
HPV, human papillomavirus; WS-A, workshop A; WS-B, workshop B; WS-C, workshop C.

introduction in Bangladesh, and to prioritise health interventions in the country.

### Strengths of the study, and comparison to findings from other studies

#### Stakeholder involvement
The MCDA process was supported by different stakeholders. Members of the different decision-making committees (NITAG), implementing bodies (EPI and others) and health professional associations were involved in every step of this study. Stakeholders of implementing agencies—EPI and CDC-DGHS—also participated in the deliberative process and ranking. NITAG members and members of National Committee for Immunization Practices also participated in the final decision-making workshop at the ministry level. Participation of stakeholders in this study ensured the transparency and accountability of decision-making, which is essential for a fair priority setting approach.[33] The importance of involving different stakeholders during the decision-making of vaccine introduction is also highlighted in other countries such as South Korea,[34] Oman,[35] Indonesia[36] and the Netherlands.[37]

#### Criteria used in priority setting
Incidence rate of the disease and case fatality rate criteria were weighted highly, indicating that disease burden was considered important for vaccine selection by the stakeholders. This finding is similar to other studies which suggest disease burden as the most common and important criterion considered by other low- and middle-income countries (LMICs) during national decision-making.[19 38–42] Efficacy of the vaccines was weighted as the next most important criterion suggesting that clinical effectiveness is also important.

#### Deliberative MCDA
The final ranking in this study was based on deliberation using the performance matrix, where the weights and scores were not explicit. Despite the lack of explicit weighting and scoring, deliberative discussions are considered to be a very important part of MCDA process as it allows a shared understanding of the data, criteria and priorities. Deliberation among stakeholders followed by simple ranking appears a feasible strategy for the prioritisation of vaccines for introduction in Bangladesh and other LMICs. Kenya and Iran selected vaccines by voting, whereas Oman, India and the Netherlands selected vaccines by expert evaluation which were evidence-based but not systematic.[35 37 43 44] Korea and Thailand selected vaccines systematically via evidence-based deliberation using DELPHI and MCDA techniques.[34 45] Recent consensus on the use of MCDA for Health Technology Assessment (HTA),[46] recommends deliberative MCDA approach over quantitative MCDA. Furthermore, a recent study by WHO encouraged weighting and scoring as they help streamline the deliberative discussions.[47] The methods used in our study, where the stakeholders deliberated the results from the quantitative MCDA and the performance matrix before finalising the ranking of vaccines, are in line with these recommendations.

### Implications for policymakers
While decision-making around vaccines in LMICs has been driven by donor funding, our study shows that it is possible to perform prioritisation systematically using evidence-based MCDA approaches. Based on the results of the MCDA study, the top-ranked JE vaccine is planned to be recommended to the decision-makers for introduction into the national vaccine benefit package. Please note that the ranking of vaccines and the selection of JE vaccine is country specific and may not be applicable to other settings. It is noteworthy that decision-making itself is a dynamic process, and vaccine performance on some criteria is likely to change over time. Therefore, we recommend Bangladesh undertake this priority setting process routinely even though most of the countries evaluate vaccines to be introduced once.[39 40 43 48–51]

### Limitations of the study
Different sets of stakeholders took part in the three workshops, resulting in a lack of a consistent group of stakeholders (and hence values/preferences) throughout the MCDA process. The ranking from quantitative weighting and scoring (from WS-A and WS-B) was different to the ranking by the stakeholders in the WS-C, who ranked the vaccines after a deliberative process. This may be due to the differences in the stakeholder membership between the different workshops and the underlying differences in their preferences.

Furthermore, the vaccine ranking in WS-C was finalised after considering the cost-effectiveness and the outbreak potentiality criteria, as well as the quantitative ranking. Also, the stakeholder preferences were implicit in the WS-C while they were explicitly elicited in the ranking using quantitative weighting and scoring (from WS-A and WS-B). This highlights the importance of ensuring a consistent set of criteria and a consistent preference

elicitation methodology throughout the MCDA process, along with a consistent group of stakeholders. If the membership, the criteria set or the methodology changes between the different workshops, there is a possibility that the ranking may change quite substantially.

Despite the inclusion of a wide variety of stakeholders, our study does not represent all stakeholders' perspectives. Stakeholders from private sectors and representatives of patient groups were not involved in the process leading to uncertainty in accountability of the results to those stakeholders.

Finally, in our study, the cost-effectiveness considerations and data of outbreak potentiality were included as qualitative criteria rather than quantitative criteria with explicit weighting and scoring. It is important to note that cost-effectiveness is not recommended as a criterion in the MCDA,[52 53] as such, a pragmatic approach was taken to consider this information qualitatively rather than weighting and scoring. While decision-making around vaccines has typically been driven by donor funding assurance, financial considerations are highlighted as being key by stakeholders. Capacity building around economic evaluation and budget impact analysis of vaccines is needed in LMICs such as Bangladesh to support evidence-based priority setting combining MCDA with Value for Money approaches.[53–55]

## CONCLUSIONS

This study presents the first application of MCDA to support vaccine prioritisation in Bangladesh health system. This study involved relevant stakeholders in priority setting process and achieved the objectives of prioritising the vaccines for introduction in Bangladesh in a transparent way, using systematic evidence-based decision-making. JE vaccine was placed as the top-ranked vaccine and is planned to be recommended to the decision-makers for introduction into the national vaccine benefit package. The use of MCDA to prioritise interventions in healthcare should be promoted as the decision-making process can be improved using systematic approaches.

**Contributors** MSH conceived and designed the study, collected data, analysed the results and drafted the manuscript. SY and PT helped in the study design, data analysis, interpretation of results and reviewed the manuscript. MT contributed to the design of the study and reviewed the manuscript. All authors discussed the findings of the study, edited and approved the manuscript. All authors are responsible for the overall content as guarantors.

**Funding** International Decision Support Initiative (iDSI) (OPP1087363)

**Disclaimer** The findings, interpretations and conclusions expressed in this article do not necessarily reflect the views of the funding agencies.

**Competing interests** The authors declared no potential conflicts of interest with respect to the research and authorship.

**Patient consent for publication** Not applicable.

**Ethics approval** Ethical clearance of this study was obtained from the Bangladesh Medical and Research Council (BMRC) and informed written consent was obtained from the stakeholders participating in the workshops.

**Provenance and peer review** Not commissioned; externally peer reviewed.

**Data availability statement** Data are available from the corresponding author. Email: sitaporn.you@mahidol.edu.

**ORCID iDs**
Mohammad Sabbir Haider http://orcid.org/0000-0003-0013-0051
Sitaporn Youngkong http://orcid.org/0000-0002-2448-3954
Montarat Thavorncharoensap http://orcid.org/0000-0002-8256-2167
Praveen Thokala http://orcid.org/0000-0003-4122-2366

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
