## [Reviewer comments · BMJ Open]

ARTICLE DETAILS

TITLE (PROVISIONAL)	Priority Setting of Vaccine Introduction in Bangladesh: A Multi-Criteria Decision Analysis Study
AUTHORS	Haider, Mohammad; Youngkong, Sitaporn; Thavorncharoensap, Montarat; Thokala, Praveen

VERSION 1 – REVIEW

REVIEWER	Farzana Muhib PATH, CVIA
REVIEW RETURNED	14-Jul-2021

GENERAL COMMENTS	There needs to be inclusion of limitations of the study that address methodologic limitations of how the MCDA is carried out, specifically the need for consistency by members of the workshops, how this exercise needs to be undertaken multiple times and whether or how it can be integrated into decision making. Also this study does not address how this methodology can be integrated into current decision making process of the NITAG, EPI and ICC. Vaccine introduction decision making in Bangladesh is, as the Authors note, complicated. With Bangladesh facing Gavi graduation, the pressure to introduce new vaccines has not only a disease burden/vaccine impact component but a financial one. It would be good to address why this was not included in the criteria. I believe that this is an incredibly timely and relevant publication and once these revisions are made it should be published.
--

REVIEWER	Hae Won Kim Seoul national university, college of nursing, the research institute of nursing science
REVIEW RETURNED	03-Sep-2021

GENERAL COMMENTS	If possible, it is desirable to specify that how many persons are involved in each workshop specifically. How to adjust or control their interest or benefit in their group in regard to make decision? Is there any document, written informed consents about participants' ethical commitment ?
---

REVIEWER	Hend Chaker Institute Pasteur of Tunisia, Laboratoire d'Histologie et de Cytogenetique
REVIEW RETURNED	25-Sep-2021

GENERAL COMMENTS	Dear authors, Here are my comments and points of concerns about the article: -The research question or study objective is clearly defined. -The abstract does not announce clearly the methodology of the research.
--

	-My comments on the methodology of the research are as follow: • The list of potential vaccines to introduce in Bangladesh, and how they were selected, as the selection of criteria for vaccine introduction, combining a review of literature and the views of regional stakeholders, as recommended by the guidelines of the ISPOR task force, on MCDA good practices, were well conducted. [1] However, I have different concerns when it comes to the methodology, mainly for how you did the scoring of the selected criteria. 1/the number of specialists/ stakeholders that have been involved in the workshops to express their preferences toward the different criteria is not mentioned. Details about the panel should be given (Number of specialists; Composition of the panel: proportion of the different specialties, virologists/ epidemiologists, etc....) 2/The methodology of the authors to score the criteria should not be explicit, using the performance matrix. Following the recommendation of the ISPOR task force, the “performance matrix” can be used, in effect, as an “aide-mémoire” for decision makers’ deliberations without explicit scoring and weighting. 3/ Weighting Criteria involves eliciting stakeholders’ preferences between criteria. The methodology that have been used by the authors to elicit their preferences between criteria was not clear. It is not said which methodology has been followed to score alternatives, and then to capture specialist preferences or priorities, to weigh up the selected criteria and levels. Using Compositional methods or decompositional methods, to score alternatives, is a required step, in MCD process [1]. In my point of view, It takes to follow a methodology to complete the rating/scoring of the criteria and the related levels, and that should not be consensually assigned as it has been done. 1. Thokala P, Devlin N, Marsh K, Baltussen R, Boysen M, Kalo Z, et al. Multiple Criteria Decision Analysis for Health Care Decision Making--An Introduction: Report 1 of the ISPOR MCDA Emerging Good Practices Task Force. Value Health. 2016;19(1):1-13.
--	--

REVIEWER	Lucia Craxi University of Palermo
REVIEW RETURNED	27-Sep-2021

GENERAL COMMENTS	Many typos to be corrected Page 5: Strengths and limitations of the study are not clearly presented.
---

VERSION 1 – AUTHOR RESPONSE

Reviewer 1’s comments to author:

Dr. Farzana Muhib, PATH

Comment:

1. There needs to be inclusion of limitations of the study that address methodologic limitations of how the MCDA is carried out, specifically the need for consistency by members of the workshops, how this exercise needs to be undertaken multiple times and whether or how it can be integrated into decision making.

Response:

We would like to thank the reviewer for the valuable comments to improve our manuscript.

We have highlighted the limitations of the methodology, as suggested, in the discussion section on page 12.

Comment:

2. Also this study does not address how this methodology can be integrated into current decision making process of the NITAG, EPI and ICC. Vaccine introduction decision making in Bangladesh is, as the Authors note, complicated. With Bangladesh facing Gavi graduation, the pressure to introduce new vaccines has not only a disease burden/vaccine impact component but a financial one. It would be good to address why this was not included in the criteria. I believe that this is an incredibly timely and relevant publication and once these revisions are made it should be published.

Response:

In this study, NITAG members and members from EPI and other organizations were involved from the beginning and in every workshop. Key decision makers of the ministry of health were involved in this study, which are mentioned in this study. ICC and ministry of finance endorse the decision of NITAG during Gavi funded vaccine introduction. We suggested to include this method in the national immunization policy, so that in the future episode of vaccine introduction this method can be used, page 9.

Donor funding is the main guidance of vaccine introduction in Bangladesh. Stakeholders express that the financial criteria is important during decision-making process, but that was overshadowed by the donor funding assurance. Bangladesh is in Gavi funding and did not evaluate financial criteria systematically, as there is lack of expertise in this field. Bangladesh will be graduating from Gavi funding soon, so there is need to develop the capacity to evaluate economic analysis, page 10. Also, please note that cost-effectiveness is not recommended as a criterion in the MCDA, 48-49 as such, a pragmatic approach was taken to consider this information qualitatively rather than weighting and scoring. Capacity building around economic evaluation and budget impact analysis on vaccines needs to be employed in LMICs such as Bangladesh to support evidence based priority setting combining MCDA with Value for Money (VfM) approaches. This issue is highlighted in the discussion section, please see page 12.

Reviewer 2's comments to author:

Dr. Hae Won Kim, Seoul national university

Comment:

If possible, it is desirable to specify that how many persons are involved in each workshop specifically. How to adjust or control their interest or benefit in their group in regard to make decision? Is there any document, written informed consents about participants' ethical commitment?

Response:

Thank you very much for the useful comments.

- We added the number of participants involved in each workshop (Workshop A 14 participants, Workshop B 6 participants, Workshop C 10 participants, and Workshop D 28 participants). This detail is now added in the manuscript.
- Before start each workshop, we discussed about the situation of vaccine introduction in the country. The participants in the workshop are members of different expert committee, NITAG members or another specific groups. We could not control the interest but let them express their own interest and exchange with the others through the workshops. So individual ranking followed by consensus among the group might make balance of their interests of any individual.
- We added 'Informed written consent was obtained from the key informants and stakeholders participated in the workshops.' to the methods section, page 5.

Reviewer 3's comments to author:

Dr. Hend Chaker, Institute Pasteur of Tunisia, Universite de Monastir Faculte de Pharmacie de Monastir

Dear authors,

Here are my comments and points of concerns about the article:

Comment:

1. The research question or study objective is clearly defined.

Response:

Thank you very much for the positive comment.

Comment:

2. The abstract does not announce clearly the methodology of the research.

Response:

The methodology of the study has been clearly added in the abstract under the design part.

Comment:

3. My comments on the methodology of the research are as follow:

The list of potential vaccines to introduce in Bangladesh, and how they were selected, as the selection of criteria for vaccine introduction, combining a review of literature and the views of regional stakeholders, as recommended by the guidelines of the ISPOR task force, on MCDA good practices, were well conducted. [1]

However, I have different concerns when it comes to the methodology, mainly for how you did the scoring of the selected criteria.

3.1 the number of specialists/ stakeholders that have been involved in the workshops to express their preferences toward the different criteria is not mentioned. Details about the panel should be given (Number of specialists; Composition of the panel: proportion of the different specialties, virologists/ epidemiologists, etc....)

Response:

Details of specialists, number of stakeholders, and number of specialists have been added to the revision, please see pages 4-7.

3.2 The methodology of the authors to score the criteria should not be explicit, using the performance matrix. Following the recommendation of the ISPOR task force, the "performance matrix" can be used, in effect, as an "aide-mémoire" for decision makers' deliberations without explicit scoring and weighting.

Response:

We followed the ISPOR MCDA taskforce good practice guidance in performing the MCDA study. We obtained the initial rankings using MCDA with quantitative weighting and scoring (from workshops A and B). This ranking was provided to the stakeholders for deliberation in workshop C. This is explained in detail below and made clearer in the manuscript.

We asked the stakeholders in workshop A (WS-A) to assign the weight and scores for the different levels of the criteria (see page 5-6). Then we provided the scores to the vaccine performance against each criterion in the performance matrix (from WS-B). From this, we obtained the first ranking of vaccines.

Then all information of performance matrix and this ranking were explicitly presented to the stakeholders in WS-C for appraisal. They reached the other rank ordering of the vaccines based on deliberation as explained in the manuscript.

3.3 Weighting Criteria involves eliciting stakeholders' preferences between criteria. The methodology that have been used by the authors to elicit their preferences between criteria was not clear. It is not said which methodology has been followed to score alternatives, and then to capture specialist preferences or priorities, to weigh up the selected criteria and levels. Using Compositional methods or decompositional methods, to score alternatives, is a required step, in MCD process [1]. In my point of view, it takes to follow a methodology to complete the rating/scoring of the criteria and the related levels, and that should not be consensually assigned as it has been done.

1Thokala P, Devlin N, Marsh K, Baltussen R, Boysen M, Kalo Z, et al. Multiple Criteria Decision Analysis for Health Care Decision Making--An Introduction: Report 1 of the ISPOR MCDA Emerging Good Practices Task Force. Value Health. 2016;19(1):1-13.

Response:

Many thanks for your observation. In this study, we followed the ISPOR MCDA good practice guidance from the report you mentioned. In this report, it is recommended that “decision makers’ deliberations focused on reaching a consensus ranking” – please see page 7 of the ISPOR MCDA Report 1.

As such, the stakeholders were asked to rank a list of criteria individually, which were then analysed and presented to the workshop before reaching consensus among them. The set of criteria were consensually approved.

Each criterion and the criteria levels were scored and weighted consensually as recommended in the ISPOR good practice guidance. We believe that consensus is necessary to take a policy decision, so we used the consensus approach for weighting and scoring. Also, please note that compositional MCDA approach was used, and direct rating methods were used to elicit the weights and scores.

Reviewer 4’s comments to author:

Dr. Lucia Craxi, University of Palermo

Comment:

1. Many typos to be corrected

Response: They were corrected.

Comment:

2. Page 5: Strengths and limitations of the study are not clearly presented.

Response:

Strengths and limitations of the study have been revised, see page 2 and discussion section of the revision.

VERSION 2 – REVIEW

REVIEWER	Farzana Muhib PATH, CVIA
REVIEW RETURNED	01-Dec-2021

GENERAL COMMENTS	Conclusion of the Abstract: Please revise statement regarding the national policy makers deciding to introduce JE, it has not been approved for introduction, but instead the NITAG has actually recommended the introduction of both JE and TCV vaccines, though even this has not been formally submitted to the govt as of Dec 1st, 2021. I would suggest making this language softer to reflect that JE vaccine is likely to be recommended or that the NITAG will be submitting its recommendation. Also, I am not aware of the govt actually agreeing to use MCDA at this point in time for decision making, i would revise again and say something a bit softer, in that the authors are confident that MCDA
---

	can be used for decision making in the future by the govt. Given the sensitivity around vaccine introduction decision making in Bangladesh i strongly recommend both the abstract and paper conclusion be modified or clarified to reflect the situation at the time of publication. If the statements above are clarified then I think the conclusions of this project are supported. I also recommend that the tools used be published in the supplemental material as this will promote use of this process in other countries.
--	---

REVIEWER	Hae Won Kim Seoul national university, college of nursing, the research institue of nursing science
REVIEW RETURNED	26-Nov-2021

GENERAL COMMENTS	Revision paper seems to be well reflected reviewers' comments. Limitations were pointed and discussed.
--

VERSION 2 – AUTHOR RESPONSE

Reviewer 1's comments to author:

Dr. Farzana Muhib, PATH

Comment:

Conclusion of the Abstract: Please revise statement regarding the national policy makers deciding to introduce JE, it has not been approved for introduction, but instead the NITAG has actually recommended the introduction of both JE and TCV vaccines, though even this has not been formally submitted to the govt as of Dec 1st, 2021. I would suggest making this language softer to reflect that JE vaccine is likely to be recommended or that the NITAG will be submitting its recommendation.

Also, I am not aware of the govt actually agreeing to use MCDA at this point in time for decision making, i would revise again and say something a bit softer, in that the authors are confident that MCDA can be used for decision making in the future by the govt.

Given the sensitivity around vaccine introduction decision making in Bangladesh i strongly recommend both the abstract and paper conclusion be modified or clarified to reflect the situation at the time of publication. If the statements above are clarified then I think the conclusions of this project are supported.

I also recommend that the tools used be published in the supplemental material as this will promote use of this process in other countries.

Response:

As advised, we have revised the conclusions of the abstract and the conclusion section of the main manuscript. Also, as suggested, we have added supplemental material describing the tools used in the study. The methods used for weighting and scoring are described in the main manuscript.

Reviewer 2's comments to author:

Dr. Hae Won Kim, Seoul national university

Comment:

Revision paper seems to be well reflected reviewers' comments.

Limitations were pointed and discussed.

Response:

Thank you very much for taking time to review our paper and for the positive comments.